# Human Placental Transcriptome Reveals Critical Alterations in Inflammation and Energy Metabolism with Fetal Sex Differences in Spontaneous Preterm Birth

**DOI:** 10.3390/ijms22157899

**Published:** 2021-07-23

**Authors:** Yu-Chin Lien, Zhe Zhang, Yi Cheng, Erzsebet Polyak, Laura Sillers, Marni J. Falk, Harry Ischiropoulos, Samuel Parry, Rebecca A. Simmons

**Affiliations:** 1Center for Research on Reproduction and Women’s Health, Department of Obstetrics and Gynecology, Perelman School of Medicine, University of Pennsylvania, Philadelphia, PA 19104, USA; ylien@pennmedicine.upenn.edu (Y.-C.L.); parry@pennmedicine.upenn.edu (S.P.); 2Division of Neonatology, Department of Pediatrics, Children’s Hospital of Philadelphia, Philadelphia, PA 19104, USA; laura.sillers@pennmedicine.upenn.edu (L.S.); ischirop@pennmedicine.upenn.edu (H.I.); 3Department of Biomedical and Health Informatics, Children’s Hospital of Philadelphia, Philadelphia, PA 19104, USA; zhangz@chop.edu; 4Mitochondrial Medicine Frontier Program, Division of Human Genetics, Department of Pediatrics, Children’s Hospital of Philadelphia, Philadelphia, PA 19104, USA; chengy1@email.chop.edu (Y.C.); polyake@email.chop.edu (E.P.); falkm@chop.edu (M.J.F.); 5Department of Obstetrics & Gynecology, University of Pennsylvania Perelman School of Medicine, Philadelphia, PA 19104, USA

**Keywords:** placenta, transcriptome, spontaneous preterm birth, bioenergetic metabolism, interactome, fetal sex disparity

## Abstract

A well-functioning placenta is crucial for normal gestation and regulates the nutrient, gas, and waste exchanges between the maternal and fetal circulations and is an important endocrine organ producing hormones that regulate both the maternal and fetal physiologies during pregnancy. Placental insufficiency is implicated in spontaneous preterm birth (SPTB). We proposed that deficits in the capacity of the placenta to maintain bioenergetic and metabolic stability during pregnancy may ultimately result in SPTB. To explore our hypothesis, we performed a RNA-seq study in male and female placentas from women with SPTB (<36 weeks gestation) compared to normal pregnancies (≥38 weeks gestation) to assess the alterations in the gene expression profiles. We focused exclusively on Black women (cases and controls), who are at the highest risk of SPTB. Six hundred and seventy differentially expressed genes were identified in male SPTB placentas. Among them, 313 and 357 transcripts were increased and decreased, respectively. In contrast, only 61 differentially expressed genes were identified in female SPTB placenta. The ingenuity pathway analysis showed alterations in the genes and canonical pathways critical for regulating inflammation, oxidative stress, detoxification, mitochondrial function, energy metabolism, and the extracellular matrix. Many upstream regulators and master regulators important for nutrient-sensing and metabolism were also altered in SPTB placentas, including the PI3K complex, TGFB1/SMADs, SMARCA4, TP63, CDKN2A, BRCA1, and NFAT. The transcriptome was integrated with published human placental metabolome to assess the interactions of altered genes and metabolites. Collectively, significant and biologically relevant alterations in the transcriptome were identified in SPTB placentas with fetal sex disparities. Altered energy metabolism, mitochondrial function, inflammation, and detoxification may underly the mechanisms of placental dysfunction in SPTB.

## 1. Introduction

Preterm birth (delivery before or at 37 weeks of gestation) is the leading cause of morbidity and mortality in newborn infants worldwide [1]. Fifteen million babies are born prematurely annually, resulting in an excess of one million deaths. Infants who survive preterm birth often have serious and lifelong health problems, including lung disease, vision loss, and neurodevelopmental disorders. Spontaneous preterm birth (SPTB) remains a significant and poorly understood perinatal complication. SPTB includes the preterm spontaneous rupture of membranes, cervical insufficiency, and preterm labor. While the exact etiology remains unknown, many factors may contribute to SPTB, including placental dysfunction, abnormal cervical remodeling, uterine distension, vascular disorders, and chorioamnionitis [2,3].

During pregnancy, the placenta facilitates nutrient transport and gas exchange and supports the growth and development of the fetus. It also produces and releases hormones into the maternal and fetal circulation to regulate uterine functions, the maternal metabolism, and fetal growth and development. Therefore, a well-functioning placenta is crucial for normal gestation. Placental dysfunction is associated with preeclampsia and fetal growth restriction. Emerging evidence suggests that placental insufficiency is also associated with a significant proportion of preterm births, especially early preterm births, as well as those complicated by chorioamnionitis [4,5,6]. The placenta protects the fetus against infections, toxins, xenobiotic molecules, and maternal diseases [7]. The placenta also produces a wide variety of metabolites, many of which are involved in energy production [8,9]. In our previous metabolomic analysis of placenta samples obtained from women with SPTB, we observed a significant elevation in the levels of amino acids, prostaglandins, sphingolipids, lysolipids, and acylcarnitines in SPTB placentas compared to term placentas [10], which suggests an imbalance between the supply capacity and metabolic demands in SPTB placentas.

Fetal sex plays an important role in pregnancy complications and perinatal outcomes. Male fetal sex is a risk factor for preeclampsia and gestational diabetes, as well as presents a higher cardiovascular and metabolic load for the mother [11,12,13,14]. A higher incidence of preterm birth is also observed among women carrying male fetuses. Although the underlying mechanism for the increased preterm birth rate of male newborns is still unclear, a potential more proinflammatory intrauterine milieu generated by male placentas may partially contribute to the increased incidences [15,16,17].

We hypothesized that deficits in the capacity of the placenta to maintain bioenergetic and metabolic stability throughout the course of pregnancy may ultimately result in SPTB. To test this hypothesis, we assessed the transcriptomes in both male and female placenta samples obtained from women with spontaneous preterm deliveries. We also integrated the transcriptomic data with our previously published metabolomic data [10] to assess the interactions of the altered genes and metabolites. Therefore, the aim of this study was to elucidate the underlying mechanisms for placental insufficiency and dysfunction, especially metabolic changes and sex differences, which will lead to a better understanding of the etiology of prematurity and the development of preventative treatments.

## 2. Results

### 2.1. Clinical Characteristics

There were no differences in maternal ages between term and spontaneous preterm birth (SPTB) placenta samples (Table 1). Of note, none of the preterm or term placental samples were from mothers with preeclampsia or gestational diabetes, and none of the women received low-dose aspirin for the prevention of preeclampsia. All of the women contributing placenta samples presented in labor with either a preterm premature rupture of membranes (PPROM), premature rupture of membranes (PROM), or cervical dilation. Thirteen women had preterm labors, and none of the women who labored at term received a betamethasone treatment prior to delivery. Three of the women with preterm labors received 17-hydroxy progesterone and five received vaginal progesterone for prematurity prevention. None of the women with term delivery received supplemental progesterone. A greater percentage of women with preterm labor received antibiotics, with a primary indication of PPROM or Group B—streptococcus (GBS) prophylaxis; however, only one woman with preterm labor was diagnosed with chorioamnionitis. Chronic medications administered to women with preterm deliveries included albuterol and inhaled corticosteroids for asthma in three women, psychotropic medications in two women, and antihypertensive drugs in four women. The chronic medications documented for women with term deliveries included albuterol in one woman and ferrous sulfate medication in one woman.

### 2.2. Global Assessment of Transcriptome Profiles in Placentas

To investigate the genes and novel pathways that are disrupted in placentas from SPTB, the gene expression profiles of placental tissues from eight male preterm cases with a mean gestational age (GA) of 29.4 weeks, seven male term controls with a mean GA of 39.7 weeks, eight female preterm cases with a mean GA of 32.1 weeks, and eight female controls with a mean GA of 39.5 weeks were assessed by RNA-Seq. The power calculation using a false-positive rate of 0.05 (two-sided) and power of 80% to target a two-fold change indicated that six samples were sufficient to determine the significant differences between the groups. One male control and one female preterm cases were considered as outliners due to the potential contamination of other cell types and were excluded from further analysis of the differential gene expressions. The principal component analysis (PCA plot, Figure 1) indicated that the transcriptome profiles for males and females were readily distinguishable. Preterm and term birth groups also separated, suggesting significant differences in the transcriptome profiles of preterm and term placentas. Genes were considered differentially expressed, with an FDR (*q*-value) ≤ 0.05. The comparison of male SPTB vs. male term deliveries yielded a total of 724 differentially expressed genes, with 347 upregulated and 377 downregulated (Figure 2a and Appendix A) (*q* ≤ 0.05 vs. term controls; *q*-value ≤ 0.05 was considered significant). Interestingly, the difference of the transcriptome profiles between female SPTB and female term deliveries was much smaller than that of the male transcriptomes. Only 66 differentially expressed genes were identified in the comparison of female SPTB vs. term, with 28 upregulated and 38 downregulated (Figure 2b and Appendix A). Most differentially expressed genes in each comparison were unique, suggesting a different molecular basis for placenta dysfunction and possibly preterm births of males vs. females. Five differentially expressed genes were identified in both male and female SPTB compared with their term controls, including four upregulated genes, *ASB4, CMAS, KATNBL1*, and *PRR9*, and one downregulated gene, *SLC28A1* (Appendix A). Differences in the expression between SPTB and term placentas of these genes are likely due to gestational age effects and unlikely to be associated with preterm births.

To determine the possible effect of gestational age (GA) on the differences in the gene expressions, we took two approaches. First, we compared our SPTB transcriptomes with the human placenta studies conducted by Eidem et al. [18] and Brockway et al. [19], who identified 37 and 170 GA-specific candidate genes, respectively. We also compared the SPTB transcriptomes with our proteomics data on four placentas from term deliveries and four placentas from elective second trimester terminations (Appendix A). In the proteomic dataset, 4711 proteins were identified, and 953 proteins were differentially expressed in term compared to second trimester placentas (proteins associated with blood were eliminated) (Appendix A). Together, we identified 54 GA-specific candidate genes in male SPTB placental transcriptome and five candidate genes in female SPTB placental transcriptome (Appendix A). These GA-specific candidate genes were removed from the datasets prior to the ingenuity pathway analysis.

### 2.3. Differences in the Transcriptome Profiles between Male and Female Placentas

Comparing male with female transcriptomes from term placentas, 319 differentially expressed genes were identified, with 177 upregulated and 142 downregulated in male compared to female placentas (Figure 2c and Appendix A). Thirty-nine differentially expressed genes were either X- or Y-chromosome-linked. In the comparison of male and female SPTB transcriptomes, in addition to 36 sex chromosome-associated genes, 144 differentially expressed genes were identified, with 105 upregulated and 39 downregulated in male compared to female STPB placentas (Figure 2d and Appendix A). These genes regulated at least 20 canonical pathways (Appendix A). Only 29 differentially expressed genes overlapped between the STPB and term groups when comparing male with female placentas. The expression of six genes, *ICAM2, AADACL3, RNR1, RNR2*, *MTNR1B*, and *HIST1H3H*, that were not associated with sex chromosomes showed clear sex differences (Table 2). The expression of *ICAM2, MTNR1B*, and *HIST1H3H* were lower in male placentas from both preterm and term births compared with female placentas. In contrast, the expression levels of *AADACL3, RNR1*, and *RNR2* were higher in male placentas.

### 2.4. IPA Identifies Multiple Pathways That Are Altered in Male SPTB Placentas

To identify the molecular pathways that may contribute to placenta dysfunction and, possibly, SPTB, an ingenuity pathway analysis (IPA) was used to map differentially expressed genes into functional networks. An IPA analysis of 670 non-GA-associated differentially expressed genes in male SPTB placentas revealed 65 canonical pathways that were altered in preterm births. As predicted by the activation z-score, the top canonical pathways activated in male SPTB included the NRF2-mediated oxidative stress response, xenobiotic metabolism pathways, and estrogen receptor signaling (Table 3). The top pathways inhibited in male SPTB were VDR/RXR activation and the antiproliferative role of the transducer of ERBB2 (TOB) in T-cell signaling (Table 3).

In addition to the identification of altered canonical pathways, IPA disease and a biological function analysis also revealed more than 65 genes that were differentially expressed in male SPTB placentas, such as *RBP4, VEGFA, EREG, PLA2G2A, CRH, PRL*, and *LEP*, which regulated the metabolic processes, particularly lipid and fatty acid metabolism (Appendix A). This was consistent with our previous metabolomics study that showed a marked elevation of multiple acylcarnitine species and significantly decreased the fatty acid oxidation in SPTB placentas [10].

#### 2.4.1. Inflammatory Signaling and Oxidative Stress Pathways Are Activated

Many environmental exposures, including infections, during pregnancy increase the production of mediators of oxidative stress and abnormal metabolism, which may lead to spontaneous preterm births [20]. Several inflammatory signaling and detoxification pathways were altered in male SPTB placentas, including xenobiotic metabolism pathways, the NRF2-mediated oxidative stress response, glutathione-mediated detoxification, superoxide radical degradation, transforming growth factor-β (TGF-β) signaling, antiproliferative role of TOB in T-cell signaling, glucocorticoid receptor signaling, and PI3K signaling (Table 3). Many of these pathways were predicted to be activated in male SPTB placentas, consistent with the presence of inflammation and oxidative stress.

Xenobiotic metabolizing enzymes and transporters play critical roles in the metabolism, elimination, and detoxification of harmful xenobiotics and toxic endogenous compounds in the placenta via nuclear receptors, including the constitutive active receptor (CAR), pregnane X receptor (PXR), and aryl hydrocarbon receptor (AHR) [21]. These three nuclear receptor superfamilies were predicted to be activated in male SPTB placentas (z-score = 2.71, 2.11, and 2.45, respectively). Twelve genes comprising these pathways were altered in male SPTB placentas, including *GSTA1, GSTM5*, *MGST1,* and *CHST2* (Appendix A).

Consistent with previous studies showing that oxidative stress is associated with SPTB [22,23], glutathione-mediated detoxification and the nuclear factor erythroid 2-like 2 (NRF2)-mediated oxidative stress response were both disrupted in male SPTB placentas. Eleven genes comprising these pathways were altered in male SPTB placentas, including *CAT, PRKCA,* and *SOD3* (Appendix A).

The IPA Tox analysis, which links gene expression to clinical pathology endpoints, further identified the genes directly contributing to mitochondrial dysfunction in male SPTB placentas, including *CAT, BTG2*, *BCL2*, and *GSTA1* (Appendix A). These genes regulate the processes such as mitochondrial transmembrane potential, permeability transition of mitochondria, depolarization of mitochondria, and reactive oxygen species detoxification.

Both TGF-β and TOB play critical roles in maintaining a normal pregnancy, and low levels of TGF-β are associated with an increased risk of preterm birth [24,25,26]. Both TGF-β and TOB signaling were predicted to be inhibited in male SPTB placentas (z-score = −0.82 and −2.00, respectively). *TGFB1*, *TGFBR1*, *INHBA*, *INHA*, and *CCNE1* were examples of differentially expressed genes comprising these pathways (Appendix A). Further, underscoring TGF-β1′s importance in SPTB placentas, its downstream signal transducers SMAD3 and SMAD4 were all predicted as inhibited regulators in male SPTB placentas. They regulate the differential expression of more than 90 genes in our dataset (Figure 3a).

Interestingly, glucocorticoid receptor signaling was altered in male, but not female, SPTB placentas. Glucocorticoids activate a number of physiologic pathways in the placenta and are also critical for fetal organ development and survival during pregnancy and parturition [27]. Previous studies have shown that term female placentas have a higher glucocorticoid receptor expression compared to term male placentas [28]. Twenty-seven genes comprising this pathway were differentially expressed, including *FGG, IL5RA, KRT24, KRT5*, and *PRL* (Appendix A).

Phosphoinositide 3-kinase (PI3K) signaling modulates the immune system during pregnancy [29]. The disruption of PI3K signaling leads to an unbalanced adaptation of the maternal innate immune system to gestation and increases the fetal mortality [30]. We found that PI3K signaling was activated in male SPTB placentas, including the differentially expressed genes *CD180*, *PIK3AP1*, and *PLCD3* (Appendix A).

Cumulatively, these data suggest that the key pathways regulating oxidative stress and the ability of the placentas to handle toxins are altered in preterm birth placentas.

#### 2.4.2. Metabolic Pathways Are Altered in SPTB Placentas

The IPA analysis identified multiple metabolic canonical pathways that were altered in male SPTB placentas, including amino acids, sphingomyelins, vitamin B6, purines, and myo-inositol (Table 3). Pathways regulating histidine, sulfite oxidation, and urea cycle metabolism were disrupted, including the differentially expressed genes *CPS1, ARG2*, and *MTHFD1* (Appendix A). Sulfite oxidation, catalyzed by mitochondrial intermembrane space enzyme sulfite oxidase, is associated with oxidative stress in the placenta.

Sphingomyelin synthase (*SGMS1*) and sphingomyelin phosphodiesterase 2 (*SMPD2)*, two genes involved in sphingomyelin metabolism, were downregulated in male SPTB placentas (Appendix A). Sphingomyelins are plasma membrane components, as well as signaling sphingolipids. An altered distribution of sphingomyelin and other sphingolipid species has been shown to play an important role in preeclampsia [31,32,33]. Sphingomyelin can also be degraded into phosphocholine and ceramide via SMPD2 [34]. Reduced levels of SMPD2 result in decreased levels of ceramides which is associated with impaired trophoblast syncytialization [35]. Ceramides also act as lipid secondary messengers and influence oxidative stress via regulating the expression and activity of antioxidant enzyme manganese-dependent superoxide dismutase (MnSOD) [36].

Pyridoxal 5′-phosphate (PLP), an active form of vitamin B6, acts as a coenzyme in the metabolism of amino acids, lipids, carbohydrates, and one-carbon units. PLP also functions as an antioxidant to prevent free radical generation and lipid peroxidation, modulates mitochondrial function, and regulates the immune system [37,38]. Five genes regulating PLP metabolism were differentially expressed in male SPTB placentas, including *DAPK1* and *PNPO* (Appendix A). These metabolic changes were consistent with our metabolomics study in SPTB placentas that metabolites of amino acids and sphingolipids were altered in the SPTB placentas [10]. Furthermore, purine metabolism is altered in male SPTB placentas, which was also observed in the placentas from an intrauterine inflammation preterm birth mouse model [39].

Several signaling pathways regulating general energy metabolism were altered in male SPTB placenta, including c-AMP-mediated signaling, signal transducer and activator of the transcription 3 (STAT3) pathway, Janus kinase 2 (JAK2) signaling, and PI3K signaling. c-AMP signaling is important for the differentiation and function of trophoblasts and placentas and is the major route to trigger trophoblast fusion [40]. It also interacts with protein kinase A (PKA) and MAPK signaling and plays a critical role in glucose and lipid metabolism [41]. c-AMP signaling was predicted to be activated in male SPTB placentas (z-score = 1.51). Twelve genes regulating the c-AMP signaling pathway were differentially expressed, including *ADCY1, PDE6H, ADORA2B*, and *CNGA1* (Appendix A). Not only is PI3K signaling key in immune system functions, it is also a key regulator of glucose and lipid metabolism and oxidative stress through modulating mitochondrial respiratory chain activity, oxidative phosphorylation, and mitochondrial integrity [42,43,44]. Via PI3K signaling, the placenta can fine-tune the supply of maternal nutrient resources to the fetus [45]. Nine genes comprising this pathway were differentially expressed (Appendix A). The PI3K complex was also predicted as an activated upstream regulator with a z-score of 2.35. It regulates expression changes of 24 genes in male SPTB placentas (Figure 3b).

These results suggest that the pathways that regulate key metabolic functions of the placenta, including fatty acids and glucose, are altered in SPTB placenta.

#### 2.4.3. Retinoids, Vitamin D, and PPAR Signaling Is Disrupted

Nuclear receptors are a superfamily of transcription factors that can bind to DNA directly and regulate the gene expression upon binding to their ligands. The IPA analysis revealed that multiple nuclear receptor signaling pathways were altered in male SPTB placentas, including retinoid X receptors (RXR), peroxisome proliferator-activated receptors (PPARs), vitamin D receptor (VDR), thyroid hormone receptor (TR), and PXR (Table 3). A total of 23 genes comprising these pathways were differentially expressed, including *IGFBP1, IL1RL1, WT1, ADCY1, TGFB1,* and *GSTA1* (Appendix A). In our study, VDR/RXR activation was predicted to be inhibited in male SPTB placentas with a z-score of −1.34, and PPARα/RXRα activation was predicted to be activated with a z-score = 1.26.

#### 2.4.4. Extracellular Matrix and Cell Adhesion Are Disrupted

The extracellular matrix (ECM) is important for the architecture of placental stroma, which supports trophoblasts and provides the environment for a healthy pregnancy. Placentas from pregnancies complicated by preeclampsia exhibit peri-villous coagulation and villous fibrosis, resulting from the overproduction of ECM in the connective tissue [46,47]. Multiple pathways regulating the extracellular matrix were altered in male SPTB placentas, including the Wnt/Ca+ pathway, fibrosis/stellate cell activation, the inhibition of matrix metalloproteases, and the epithelial–mesenchymal transitional pathway (Table 3). Thirty-two genes comprising these pathways were differentially expressed, including *PDE6H, COL11A2, COL24A1, LEP, IL1RL1, PROK1,* and *MMP12* (Appendix A).

Cell–cell and cell–ECM communication are important in coordinating proliferation and differentiation during placenta development. Cell adhesion molecules, including transmembrane receptor integrins, can facilitate cell–cell and cell–ECM adhesion [48,49]. Pathways regulating cell adhesion were altered in male SPTB placentas, including epithelial adherens junction signaling, integrin signaling, and gap junction signaling (Table 3). Twenty-five genes comprising these pathways were differentially expressed, including *NOTCH3*, *ITGAD*, *ITGB6*, *MYLK3*, *ADCY1*, and *HTR2B* (Appendix A).

#### 2.4.5. Estrogen Receptor Signaling Is Activated

Interestingly, estrogen receptor signaling was predicted to be activated in male SPTB placentas with a z-score of 1.89 but was unaltered in female SPTB placentas (Table 3). Eighteen genes comprising this pathway were differentially expressed, including *ADCY1*, *LEP*, *MMP12*, *PROK1*, and *VEGFA* (Appendix A). Estrogen receptor signaling plays a critical role in trophoblast differentiation, placental function, and fetal development and modulates placenta and fetal communication [50,51,52]. Estrogen also regulates placental angiogenesis via modulating the expression of VEGF, angiopoietin-1, and angiopoietin-2 [53]. Estrogen receptor alpha and beta expression in the placenta are altered in preeclampsia and intrauterine growth restriction [54].

### 2.5. Canonical Pathways Altered in Female SPTB Placentas

An IPA analysis of 61 non-GA-associated differentially expressed genes in female SPTB placentas revealed nine canonical pathways that were altered in preterm births (Table 4). Most of these are critical pathways regulating the metabolism and nutrient sensing, including PKA signaling, insulin-like growth factor 1 (IGF-1) signaling, G-protein-coupled receptor (GPCR)-mediated nutrient sensing, α-adrenergic signaling, and cholecystokinin/gastrin-mediated signaling. PKA pathway activation plays a major role in steroidogenic gene regulation in human placentas [55]. PKA is also located in mitochondria and regulates mitochondrial protein phosphorylation [56]. IGF-1 plays a critical role in fetal and placenta growth and development [57,58]. Interestingly, maternal IGF-1 and IGF1R polymorphisms are associated with preterm birth [59] and the expression of IGF binding proteins (IGFBPs) are altered in placentas from idiopathic spontaneous preterm births [21]. α1-Adrenergic signaling stimulates the placenta blood flow, and dysregulation of this pathway has been implicated in placenta ischemia-induced hypertension [60]. Finally, cholecystokinin is one of the most highly upregulated genes in early placentas from women who later developed preeclampsia compared with women who experienced a normal pregnancy; however, its role in the placenta has not been investigated [61]. *CHD5*, *GH1*, *PRKAR1A*, *HIST1H1A*, *PYGM*, *YWHAZ*, and *CCK* were the differentially expressed genes regulating these pathways (Appendix A).

The IPA disease and biological function analysis also revealed that 17 genes in female SPTB placentas that regulate lipid, protein, and carbohydrate metabolism, including *CCK*, *GH1*, *APOC3*, *SULT1E1*, *TGM3*, and *ASB4* (Appendix A), were differentially expressed.

Overall, our data indicate that nutrient sensing and lipid, protein, and carbohydrate metabolism are disrupted in female SPTB placentas.

### 2.6. Upstream Regulators and Regulatory Networks Regulating Nutrient-Sensing, Metabolic, and Mitochondrial Function Are Altered in SPTB Placenta

The ingenuity pathway analysis can identify upstream regulators that mediate changes in the gene expression. The top upstream regulators identified in male SPTB placenta are listed in Table 5. The activated regulators in male SPTB placentas include SMARCA4, RAF1, and JUN. The important inhibited regulators include PHB2, α-catenin, TP63, and CDKN2A. Many of these upstream regulators are involved in glucose and lipid metabolism, nutrient-sensing, and mitochondrial function. SMARCA4 is part of the large ATP-dependent chromatin remodeling complex SNF/SWI. It regulates the transcription of many genes for fatty acid and lipid biosynthesis [62,63]. In addition, SMARCA4 also plays a role in trophoblast stem cell renewal and placenta development [64]. SMARCA4 was predicted as an activated regulator (z-score = 3.20) and modulated the expression changes of 28 genes in male SPTB placentas (Figure 3c). Oncoprotein c-Jun (JUN) regulates the cell cycle and apoptosis, and the expression is altered in the placenta in pregnancies complicated by preeclampsia [65]. JUN was predicted as an activated regulator (z-score = 2.21) and regulates expression changes in 31genes in male SPTB placentas. Cyclin-dependent kinase inhibitor 2A (CDKN2A) regulates cell senescence and was predicted as an inhibited upstream regulator, and the expression of 15 of its downstream targets was disrupted (Figure 3d). Attenuation of the senescence program occurs in IUGR human placentas, and a knockdown of the CDKN2A expression results in functional and morphological abnormalities in murine placenta [66]. Tumor protein p63 (TP63) was predicted as an inhibited regulator (z-score = −2.01), modulating the expression changes of 25 genes in male SPTB placentas (Figure 3e). It regulates cytotrophoblast differentiation and fusion [67].

In female SPTB placentas, the important upstream regulators identified by IPA are listed in Table 6. Insulin (INS) was predicted as an inhibited upstream regulator in female SPTB placentas with a z-score = −1.09. Phosphatase and tensin homolog (PTEN), an inhibitor of PI3K signaling, was also predicted to be inhibited with a z-score of −1, suggesting that PI3K signaling was also activated in female SPTB placentas. Vascular endothelial growth factors (VEGFs) were predicted as inhibited upstream regulators with a z-score = −1.09. They are important for angiogenesis and vascular remodeling, a process critical for placental function and healthy pregnancies.

The IPA analysis can expand predictions and identify the potential novel master regulators responsible for the changes in the gene expressions. LGALS1, NOX1, DNAJA3, PPP2R2A, PPP2CA, and BRCA1 were a few examples of master regulators identified in male SPTB placentas (Table 7). Galectin 1 (LGALS1) is a β-galactoside-binding protein and regulates maternal–fetal immune tolerance and maintaining a normal pregnancy [68,69,70]. The dysregulation of LGALS1 expression is associated with preeclampsia [71], and interestingly, maternal serum LGALS1 levels are significantly higher in pregnancies with premature ruptures of the membranes [72]. LGALS1 was predicted as an activated master regulator in male SPTB placenta and interacts with 15 upstream regulators to regulate the gene expression. NADPH oxidase 1 (NOX1), a member of the NADPH oxidase family, was predicted as an activated master regulator in male SPTB placenta. NADPH oxidase is the major source of superoxide in placentas and plays a role in early placental development [73,74]. However, the overexpression of NOX1 increases oxidative stress and is associated with preeclampsia [75]. DnaJ heat shock protein family member A3 (DNAJA3) is a mitochondrial protein regulating protein folding, degradation, and complex assembly. It plays a critical role in maintaining the mitochondrial membrane potential and mitochondrial DNA integrity; however, its role in the placenta is unknown [76]. DNAJA3 was predicted to be inhibited in SPTB placentas (z-score = −3.16) and interact with 11 upstream regulators to regulate the gene expression. Both PPP2R2A, a regulatory subunit of protein phosphatase 2 (PP2A), and PPP2CA, the catalytic subunit of PP2A, were predicted as inhibited master regulators in male SPTB placentas with a z-score of −3.03 and −1.51, respectively. PP2A acts as a negative regulator of cell growth and division and controls energy metabolism and redox homeostasis via modulating AMP kinase (AMPK) and PI3K-AKT-mTOR signaling [77]. AMPK is a master metabolic regulator controlling glucose sensing and uptake, lipid metabolism, glycogen, cholesterol, and protein synthesis, and the induction of mitochondrial biogenesis [78,79]. mTOR regulates energy-sensing pathways and functions as an important placental growth signaling the sensor to regulate trophoblast proliferation [80,81,82,83]. Furthermore, PP2A can control genome integrity by coupling the metabolic processes with DNA damage responses [84]. The breast cancer type 1 susceptibility protein (BRCA1), a tumor suppressor, regulates the cell cycle and DNA damage repair in the placenta [85]. It also functions as a regulator of glucose and lipid metabolism, as well as mitochondrial respiration [86,87,88]. BRCA1 was predicted as an inhibited master regulator in the male SPTB placentas (z-score = −2.08) and interacts with 27 upstream regulators to modulate the gene expression.

NFAT and CERK are master regulators that were identified in female SPTB placentas (Table 8). Nuclear factors of activated T cells (NFAT) is a family of transcription factors that activate cytokine production and also positively regulate placental FLT-1 and sFlt-1 e15a, the secretion of sFlt-1, and the inflammatory cytokine expression [89]. NFAT is thought to be involved in the pathophysiology of preeclampsia. In addition, NFAT may act as a regulator for the parturition and induction of labor [90]. NFAT was predicted as an activated master regulator and interacted with 23 upstream regulators to regulate the gene expression. Ceramide kinase (CERK) converts ceramide to ceramide-1-phosphate. CERK was predicted as an inhibited master regulator in female SPTB placentas and interacting with 13 upstream regulators to regulate the gene expression.

In summary, these results demonstrate that upstream regulators and master regulators important for the nutrient-sensing and metabolism are altered in SPTB placentas.

### 2.7. Interactome Network Analysis of the Transcriptome and Metabolome in SPTB Placentas

To further investigate the association of differentially expressed genes and significantly changed metabolites, identified in our previous metabolomics study [10]; an interactome network model (Figure 4) integrating the transcriptome and metabolome was generated that was connected via protein–protein or protein–metabolite interactions. Since the placentas from both sexes were used in our metabolomics study [10], we used the transcriptomes from the combined sexes (Appendix A) for the interactome network analysis. The analysis of this interactome network confirmed that several metabolic processes are altered in SPTB placentas. These modules included the genes and/or metabolite interactions that were associated with fatty acid metabolism, cholesterol biosynthesis, steroid hormone metabolism, glycolysis and gluconeogenesis, pentose phosphate pathway, TCA cycle, amino acid metabolism, purine and pyrimidine metabolism, glycosphingolipid metabolism, glycerophospholipid metabolism, phosphatidylinositol phosphate metabolism, vitamin metabolism, and xenobiotics metabolism (Table 9). The differentially expressed genes and significantly altered levels of metabolites associated within these modules in our datasets are also listed in Table 9.

## 3. Discussion

In this study, we demonstrated marked changes in the expression of genes in SPTB placentas involving key pathways regulating mitochondria function, inflammation, amino acid and lipid metabolism, extracellular matrix, and detoxification. Importantly, we also show that there are marked differences in the placental transcriptome in SPTB between males and females, suggesting that there may be differences between males and females in the mechanisms by which a placenta dysfunction contributes to SPTB.

Male fetuses have a higher incidence of many pregnancy complications, including preterm births [15,16,17]. Preterm males also have increased morbidity and mortality after births [91]. Although the underlying mechanisms are unclear, a more proinflammatory intrauterine milieu at lower gestational ages may account for the increased incidence and/or make the male fetus more susceptible to an abnormal intrauterine milieu. In the current study, 670 differentially expressed genes were identified in male SPTB compared to term placentas. To our surprise, only 61 differentially expressed genes were found in female SPTB placentas, supporting the observations of fetal sex differences and the lower susceptibility to spontaneous preterm birth in female fetuses. The limited changes of the transcriptomes in female SPTB placentas compared with term placentas may result in a survival advantage for females and adaptive responses for a suboptimal milieu, as previously reported [92,93].

We identified multiple metabolic pathways that were altered in the SPTB placenta in our previous metabolomics study [10]. Levels of sphingolipids, steroids, amino acids, and metabolites involved in fatty acid oxidation, such as acylcarnitines, were significantly different between SPTB and term placentas. Acylcarnitines, the major metabolites increased in SPTB placentas, are intermediate oxidative lipids and are associated with proinflammatory signaling and mitochondrial dysfunction [94,95]. Multiple elevated 2-hydroxy long-chain fatty acids in SPTB placentas are potent uncouplers of oxidative phosphorylation and can impair energy homeostasis and induce the mitochondria permeability transition pore. The current study identifies the molecular basis for these changes as multiple genes, and the pathways controlling these processes were substantially altered. The IPA analysis of the transcriptome data demonstrated that many pathways and upstream regulators regulating inflammation, mitochondrial function, redox status and signaling, and energy metabolism and homeostasis were significantly altered in SPTB placentas. The alteration of these pathways suggests a fundamental disruption of mitochondria metabolism, as well as the initiation of a proinflammatory milieu in SPTB, and an activation of oxidative response/detoxification pathways may reflect an adaptive response, which ultimately fails, resulting in SPTB. In fact, our current findings were similar to the observations in our previous study with an intrauterine inflammation preterm birth mouse model [39], supporting that mitochondria dysfunction, abnormal fatty acid, and inflammation play major roles in SPTB even in the absence of overt infections.

Our finding that glucocorticoid receptor signaling was altered in male, but not female, SPTB placentas was intriguing. Glucocorticoids are critical for implantation, fetal organ development, and survival during pregnancy and parturition [27]. However, excess glucocorticoid exposure suppresses the immune system and has adverse effects on placental proliferation, angiogenesis, and glucose transport [28,96,97]. The observation that a higher glucocorticoid receptor expression in term female placentas compared to term male placentas suggests the lower glucocorticoid exposure of female fetuses during pregnancy and an enhanced immune response, which may contribute to the increased survival rate of female fetuses in an aberrant intrauterine milieu [28]. The effects of glucocorticoids on the placentas and fetuses also show a sex-specific manner. Glucocorticoid exposure increases oxidative stress in the male placentas but not the female placentas [98]. Glucocorticoids also decrease the adrenal activity in preterm males but not females [99], which may partially account for the poor perinatal outcomes of preterm males.

Other pathways that were altered in SPTB placentas include retinoids, vitamin D metabolism/signaling, thyroid hormone, and PPARs. Retinoids are lipophilic molecules and metabolites of Vitamin A (all-trans-retinol). They play important roles in regulating the energy metabolism and function as critical regulators during embryogenesis and promote the differentiation of trophoblast stem cells [100,101,102]. The actions of retinoids are mediated through retinoic acid receptors (RARs) and RXRs [103]. RXRs are common heterodimer partners for multiple nuclear receptors, such as PPARs, VDR, TR, and PXR [104]. Vitamin D plays a critical role in pregnancy in additional to its classical role in calcium/phosphate homeostasis and bone metabolism. It regulates he decidualization and implantation, hormone secretion, and placental immune response and defended the infections [105,106]. Vitamin D also has a potent antioxidant effect to prevent protein oxidation, lipid peroxidation, DNA damage, and maintaining a normal mitochondrial function [107]. The deficiency of vitamin D is associated with impaired fetal growth, preeclampsia, and gestational diabetes [105]. Indeed, epidemiologic studies also provide the evidence linking vitamin D insufficiency with preterm births [108]. The thyroid hormone (TH) plays a critical role in regulating the metabolic processes for normal growth. It can regulate gene expression directly, as well as crosstalk with PPAR and the liver X receptor (LXR), and modulate glucose, lipid, and cholesterol metabolism [109]. TH is important for the healthy pregnancy modulating for cell proliferation and differentiation, metabolism, and formation and functioning of the placenta [110]. It may play an important role in fine-tuning inflammation in placentas in both term and preterm labors [111]. The dysregulation of maternal thyroid hormone signaling is associated with preeclampsia, miscarriage, and intrauterine growth restriction [110]. In addition to drug transport, PXR/RXR heterodimer regulates the homeostasis/metabolism of glucose, lipid, steroids, bile acids, retinoic acid, and bone minerals [112]. PPARs modulate the inflammatory responses, cell proliferation, and cell division. They also play a central role in placental angiogenesis [113,114,115,116,117]. PPARs also exert the antioxidant effects and are critically important to early placental development [118,119].

Multiple pathways regulating the extracellular matrix and cell adhesion were disrupted in SPTB placentas. ECM is important for the architecture of placental stroma and supports a healthy pregnancy. Of note, ECM also plays an important role in the nutrient uptake, redox status, energy metabolism, and mitochondrial function. The ECM also regulates glucose transport, glycolysis, lipid metabolism, and the TCA cycle [120,121,122]. AMPK, the master metabolic regulator, also regulates integrin activity and extracellular matrix assembly [123]. ECM remodeling can modulate mitochondrial structure, dynamics, and function [124].

Integrated interactome modeling provides greater confidence in the signaling and pathways identified in the transcriptome and metabolome individually. Alterations of the lipid metabolism in SPTB placentas were identified in all three analyses of the transcriptome, metabolome, and interactome, which further supports that aberrant fatty acid metabolism may be causal to preterm birth. Interactome modeling also showed that metabolism, for almost all amino acids, glucose, steroid hormones, purines, pyrimidines, and vitamins, were altered in SPTB placentas. Collectively, our results strongly suggest that alterations and/or deficits in metabolic pathways cause placental insufficiency, ultimately resulting in SPTB.

A major limitation in spontaneous preterm birth research is the lack of human gestational controls. Eidem et al. used Rhesus macaque as gestational age controls in their study and identified 267 differentially expressed genes between preterm and term human placentas, including 29 SPTB-specific candidate genes [18]. These “SPTB” genes are enriched for functions in the metabolism, immunity, inflammation, and cell signaling, which are consistent with the results in our current study. Brockway et al. used infections related to preterm birth as a gestational age control and identified 170 SPTB-specific genes [19]. Similarly, these genes are also enriched for pathways in insulin-like growth factor (IGF) signaling, cytokine signaling, immune system, and signal transduction. While it is not a perfect control, we used second trimester placenta as a gestational age control in the current study. We also adapted the approach of Eidem et al. [18] and demonstrated that at least the differences of acylcarnitine metabolites, the major changes in the metabolome, between the Rhesus macaque preterm and term placentas, are not caused by the difference in gestational ages [10].

A well-functioning placenta plays a crucial role in normal pregnancy. Our current study has identified alterations in novel pathways and upstream regulators that may play an important role in the maintenance of normal bioenergetic metabolism and provides new insights into the underlying mechanisms of SPTB. Larger studies in preterm birth will be necessary to determine whether these findings can be generalized beyond the African American population that was studied and the possible population disparities.

## 4. Materials and Methods

### 4.1. Clinical Characteristics

Placenta samples from Black women (self-identified race) in the current study were selected from the larger Cellular Injury and Preterm Birth (CRIB 821376, NCT02441335) study at the University of Pennsylvania. CRIB enrollment criteria included women aged 18–45 years with singleton pregnancies admitted to the hospital with either spontaneous labor (defined as regular contractions and cervical dilation) or the premature rupture of membranes (PROM) occurring between 20 0/7 and 36 6/7 weeks of gestational age (preterm) or at 38 to 41 weeks of gestational age (term). The CRIB exclusion criteria included multiple gestations, fetal chromosomal abnormalities, major fetal anomalies, intrauterine fetal demise, intrauterine growth restriction, clinical chorioamnionitis, induction of labor, elective cesarean delivery, gestational diabetes, and gestational hypertension or preeclampsia. The CRIB study was approved by the Institutional Review Board at the University of Pennsylvania (protocol #821376), and patients were enrolled after written informed consent.

Second trimester placenta samples from Black women (self-identified race) were selected from the “*Trophoblast cells Isolation from Second and first Trimester placenta*” (TrISecT) study at the University of Pennsylvania and were utilized as gestational age controls in the current study. The enrollment criteria included women aged 18–45 years receiving care at the hospital due to the elective termination of a singleton pregnancy prior to 23 6/7 weeks of gestational age. The TrISecT exclusion criteria included multiple gestation, aneuploidy, and fetal congenital anomalies. The TrISecT study was approved by the Institutional Review Board at the University of Pennsylvania (protocol #827072), and patients were enrolled after written informed consent.

### 4.2. Total RNA Isolation and RNA-Seq Library Preparation

Placenta samples were collected from mid-placenta near the cord insertion on the fetal side and flash-frozen at the time of delivery (within 10 min) and stored at −80 °C prior to RNA extraction. Total RNA was extracted using TRIzol^®^ Reagent (Invitrogen, Waltham, MA, USA), followed by Qiagen RNeasy^®^ Mini Columns (Qiagen, Germantown, MD, USA) following the manufacturer’s instructions. RNA integrity numbers greater than 7 were used for RNA-Seq Studies. RNA-Seq libraries were generated, using the Agilent SureSelect strand-specific RNA library preparation kit (Agilent, Santa Clara, CA, USA).

### 4.3. RNA-Seq and Gene Expression Analysis

RNA-Seq libraries were paired-end sequenced to 100 bp on an Illumina Hi-Seq platform in CAG Sequencing Core at the Children’s Hospital of Philadelphia. RNA-seq data in .fastq files were aligned to the reference human genome (hg38) and transcriptome using the STAR (https://github.com/alexdobin/STAR, accessed on 10 June 2021) program in 2-pass mode. The alignment results were saved as indexed .bam files. Aligned reads in .bam files were loaded into R and mapped to known genes. Read pairs uniquely mapped to the sense strand of that transcription were counted to obtain a gene-level read count matrix. Differential gene expression was tested by DESeq2. Differential gene expression was evaluated by the fold change, DESeq2 *p*-value, and corresponding false discovery rate (FDR). Sequence data were deposited in NCBI’s Gene Expression Omnibus and are accessible through the GEO Series accession number GSE174415. A functional analysis was conducted using QIAGEN’s Ingenuity^®^ Pathway Analysis (IPA^®^) (Qiagen, Germantown, MD, USA). Core analyses were performed on genes with FDR (*q*-value) < 0.05.

### 4.4. Sample Preparation for Proteomics

Frozen placenta samples from normal term births, and the second trimester (*n* = 4 for each group) were sent to the Proteomics Core Facility at the Children’s Hospital of Philadelphia for protein hydrolysis, followed by peptide separation, and analyzed by LC-MS/MS on a QExactive HF mass spectrometer (Thermo Fisher Scientific, San Jose, CA, USA) coupled with an Ultimate 3000. A label-free approach was chosen for its adaptability to include new samples when needed, as well as to avoid the possible errors while the labeling techniques were applied.

### 4.5. Protein Sequence Database Search and Proteomics Analysis

MS/MS raw files were searched against a human protein sequence database, including isoforms from the UniProt Knowledgebase (taxonomy:10090 AND keyword: “Complete proteome (KW-0181)”), using MaxQuant [125] version 1.6.1.0 with the following parameters: fixed modifications, carbamidomethyl (C); decoy mode, revert; MS/MS tolerance, FTMS 20 ppm; False Discovery Rate (FDR) for both peptides and proteins of 0.01; minimum peptide length of 7; modifications included protein quantification, acetyl (protein N-term), and oxidation (M); peptides used for protein quantification, razor, and unique. iBAQ values were used for protein quantification.

Perseus (1.6.1.1) was used for proteomic data processing and statistical analysis. Protein groups containing matches to decoy database or contaminants were discarded. The data were Log_2_-transformed and normalized by subtracting the median for each sample. Student’s *t*-test was employed to identify differentially expressed proteins. Benjamini–Hochberg correction was applied to obtain an FDR.

### 4.6. Integrated Network Analysis of the Transcriptome and Metabolome

Differentially expressed genes and significantly changed metabolites identified in our previous study [10] were analyzed using MetScape3.1 [126] in Cytoscape (v3.8.0). The interactome networks were generated based on known protein–protein and protein–metabolite interactions. The metabolic pathways that were associated with protein–metabolite interactions were mapped onto each network.

## Figures and Tables

**Figure 1 ijms-22-07899-f001:**
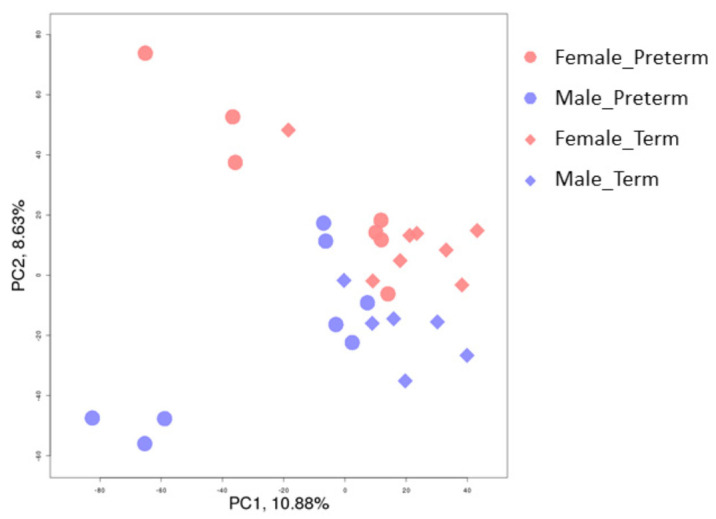
Principal component analysis (PCA plot) of the placental transcriptomes. The PCA plot revealed a significant separation between the male and female placentas, as well as the placentas from preterm and term births.

**Figure 2 ijms-22-07899-f002:**
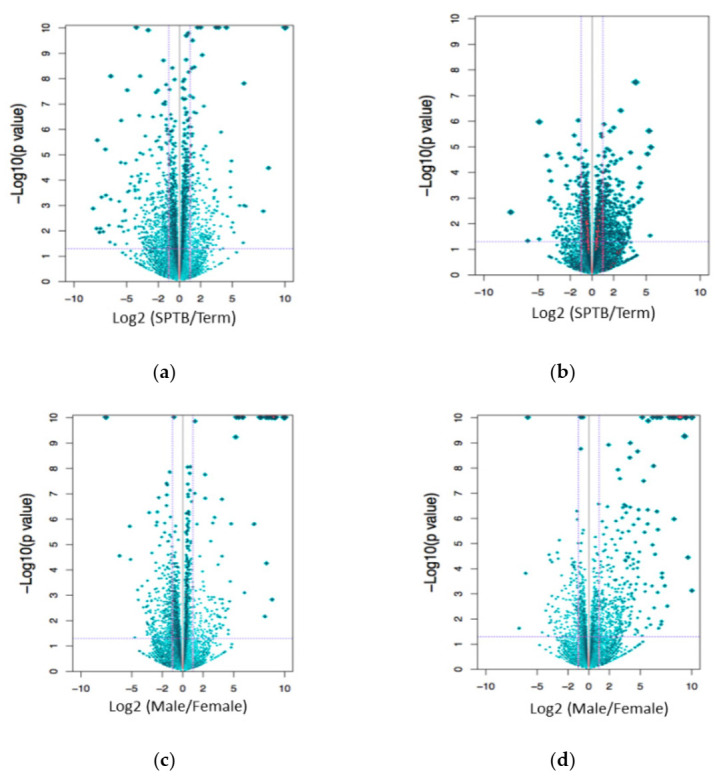
Volcano plots identifying differentially expressed genes with an FDR (*q*-value) < 0.05. Male SPTB placentas compared with term placentas (**a**). Female SPTB placentas compared with term placentas (**b**). Male term placentas compared with female term placentas (**c**). Male SPTB placentas compared with female SPTB placentas (**d**).

**Figure 3 ijms-22-07899-f003:**
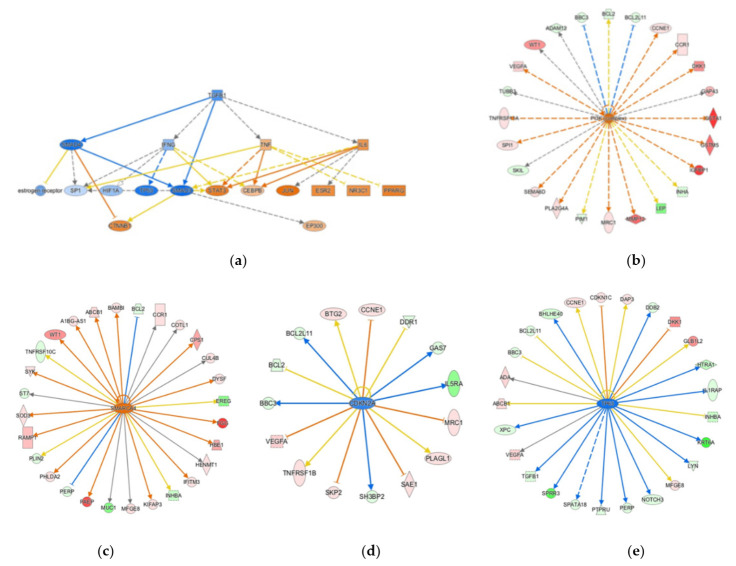
Ingenuity Pathway Analysis^®^ (IPA)-annotated mechanistic network or differentially expressed genes regulated by critical upstream regulators. Mechanistic network regulated by TGFB1, SMAD3, and SMAD4 (**a**). Differentially expressed genes regulated by the PI3K complex (**b**), SMARCA4 (**c**), CDKN2A (**d**), and TP63 (**e**). Orange-filled and blue-filled shapes indicate predicted activation and inhibition, respectively; red-filled and green-filled shapes indicate increased and decreased expressions, respectively; orange-red lines indicate activation; blue lines indicate inhibition; yellow lines indicate findings inconsistent with the state of downstream activity; grey lines indicate that the effect was not predicted.

**Figure 4 ijms-22-07899-f004:**
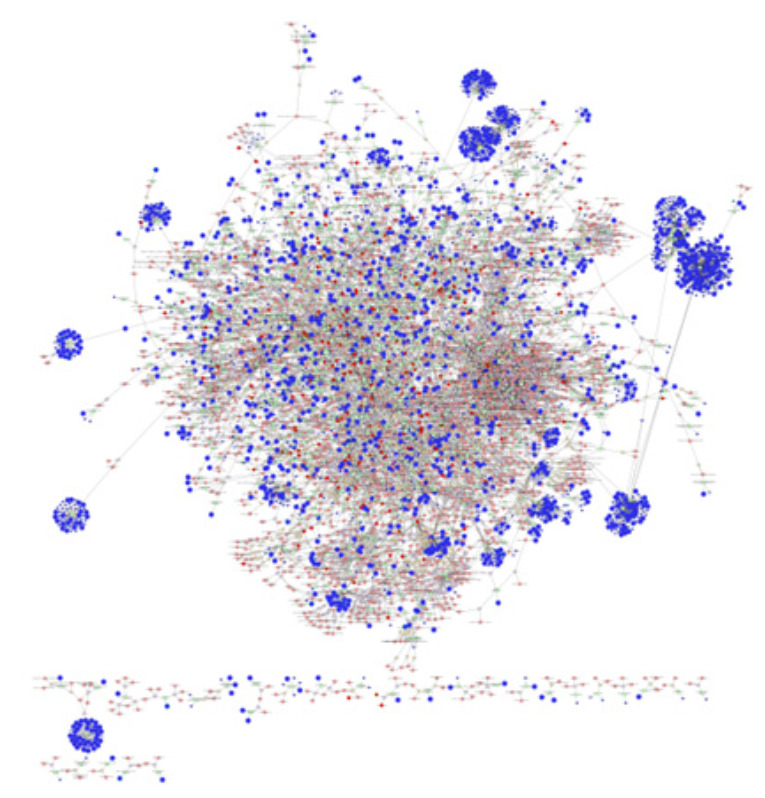
Visual representation of the interactome model. Interaction network of integrated transcriptome and metabolome was analyzed using MetScape 3.1. Dark blue circles represent differentially expressed genes in the placenta dataset; light blue circles represent inferred gene interactions; dark red circles represent significantly changed metabolites in the placenta dataset; light red circles represent inferred metabolite interactions; grey lines represent protein–protein or protein–metabolite interactions.

**Table 1 ijms-22-07899-t001:** Demographics of the study cohort.

	Preterm_Male	Term_Male	*p*-Value	Preterm_Female	Term_Female	*p*-Value
	(*n* = 8)	(*n* = 7)		(*n* = 8)	(*n* = 8)	
Gestational age at birth wks, mean ± SD	29.4 ± 4.3	39.7 ± 0.7	*p* < 0.0001	32.1 ± 4.0	39.5 ± 0.7	*p* = 0.0001
Maternal age at delivery yrs, mean ± SD	28.4 ± 5.5	28.6 ± 4.4	*p* = 0.94	26.3 ± 4.1	27.9 ± 6.2	*p* = 0.55
Maternal BMI at first visit Kg/m^2^, mean ± SD	33.4 ± 13.2	25.6 ± 4.4	*p* = 0.17	28.3 ± 5.6	28.9 ± 6.9	*p* = 0.85
Mode of delivery, n (%) Vaginal	3 (38)	6 (86)	ND	8 (100)	5 (62)	ND
C-section	5 (62)	1 (14)		0 (0)	3 (38)	
Fetal growth						
restriction, n (%)	1 (13)	0 (0)	ND	0 (0)	0 (0)	ND
Antibiotic administration						
Yes, *n* (%)	6 (75)	0 (0)	ND	4 (50)	2 (25)	ND

ND: not determined.

**Table 2 ijms-22-07899-t002:** Overlay of differentially expressed genes comparing male and female placentas at both SPTB and term births.

Gene	Gene Names	SPTB_Male_vs._Female LogFC	FDR	Term_Male_vs._Female LogFC	FDR
*RNR2*	l-rRNA	2.83	9.30 × 10^−6^	2.20	9.10 × 10^−5^
*AADACL3*	arylacetamide deacetylase like 3	5.72	7.50 × 10^−4^	3.14	3.80 × 10^−4^
*RNR1*	s-rRNA	2.55	3.50 × 10^−3^	1.95	1.50 × 10^−2^
*HIST1H3H*	histone cluster 1 H3 family member h	−1.39	1.40 × 10^−2^	−0.70	4.90 × 10^−2^
*MTNR1B*	melatonin receptor 1B	−1.40	1.60 × 10^−2^	−2.02	2.90 × 10^−2^
*ICAM2*	intercellular adhesion molecule 2	−0.75	1.60 × 10^−2^	−0.76	3.80 × 10^−2^

**Table 3 ijms-22-07899-t003:** Top canonical pathways altered in male SPTB placentas.

Ingenuity Canonical Pathways	*p*-Value	z-Score
STAT3 Pathway	6.46 × 10^−5^	0.63
Glucocorticoid Receptor Signaling	8.71 × 10^−5^	——
Adenine and Adenosine Salvage III	5.25 × 10^−4^	——
LPS/IL-1 Mediated Inhibition of RXR Function	5.50 × 10^−4^	−0.38
Wnt/Ca+ pathway	5.50 × 10^−4^	0.00
Guanine and Guanosine Salvage I	9.12 × 10^−4^	——
Inhibition of Matrix Metalloproteases	1.07 × 10^−3^	0.82
Fibrosis / Stellate Cell Activation	1.70 × 10^−3^	——
Glutathione-mediated Detoxification	2.57 × 10^−3^	——
Myo-inositol Biosynthesis	8.71 × 10^−3^	——
TGF-β Signaling	8.91 × 10^−3^	−0.82
Estrogen Receptor Signaling	1.17 × 10^−2^	1.89
Urea Cycle	1.26 × 10^−2^	——
NRF2-mediated Oxidative Stress Response	1.29 × 10^−2^	1.34
TR/RXR Activation	1.38 × 10^−2^	——
PXR/RXR Activation	1.38 × 10^−2^	——
PPARα/RXRα Activation	1.41 × 10^−2^	1.26
Regulation of the Epithelial-Mesenchymal Transition Pathway	1.45 × 10^−2^	——
Pyridoxal 5′-phosphate Salvage Pathway	1.48 × 10^−2^	0.45
Purine Ribonucleosides Degradation to Ribose-1-phosphate	1.74 × 10^−2^	——
Epithelial Adherens Junction Signaling	1.78 × 10^−2^	——
Role of JAK2 in Hormone-like Cytokine Signaling	1.91 × 10^−2^	——
Histidine Degradation III	2.29 × 10^−2^	——
Sphingomyelin Metabolism	2.29 × 10^−2^	——
Superoxide Radicals Degradation	2.29 × 10^−2^	——
PI3K Signaling	2.51 × 10^−2^	0.33
Sulfite Oxidation IV	3.02 × 10^−2^	——
Xenobiotic Metabolism CAR Signaling Pathway	3.02 × 10^−2^	2.71
Integrin Signaling	3.02 × 10^−2^	0.30
Antiproliferative Role of TOB in T Cell Signaling	3.02 × 10^−2^	−2.00
Xanthine and Xanthosine Salvage	3.02 × 10^−2^	——
VDR/RXR Activation	3.09 × 10^−2^	−1.34
Superpathway of D-myo-inositol (1,4,5)-trisphosphate Metabolism	3.16 × 10^−2^	——
Xenobiotic Metabolism PXR Signaling Pathway	3.31 × 10^−2^	2.11
BEX2 Signaling Pathway	3.31 × 10^−2^	1.63
Gap Junction Signaling	3.98 × 10^−2^	——
Xenobiotic Metabolism AHR Signaling Pathway	4.47 × 10^−2^	2.45
cAMP-mediated signaling	4.68 × 10^−2^	1.51

**Table 4 ijms-22-07899-t004:** Top canonical pathways altered in female SPTB placentas.

Ingenuity Canonical Pathways	*p*-Value	z-Score
PPARα/RXRα Activation	1.38 × 10^−2^	——
Protein Kinase A Signaling	2.09 × 10^−2^	1.00
α-Adrenergic Signaling	2.63 × 10^−2^	——
NER Pathway	3.02 × 10^−2^	——
IGF−1 Signaling	3.09 × 10^−2^	——
GPCR-Mediated Nutrient Sensing in Enteroendocrine Cells	3.55 × 10^−2^	——
Cholecystokinin/Gastrin-mediated Signaling	3.98 × 10^−2^	——
Inhibition of ARE-Mediated mRNA Degradation Pathway	4.17 × 10^−2^	——
Nitric Oxide Signaling	4.68 × 10^−2^	——

**Table 5 ijms-22-07899-t005:** Top upstream regulators altered in male SPTB placentas.

Regulators	*p*-Value	Activation z-Score	# Genes Regulated
SMARCA4	1.66 × 10^−2^	3.20	28
RAF1	5.01 × 10^−4^	2.40	16
JUN	1.83 × 10^−4^	2.21	31
IL33	1.85 × 10^−3^	2.20	16
IL13	1.00 × 10^−6^	2.19	33
SOX7	1.06 × 10^−3^	2.12	6
LIF	1.31 × 10^−3^	2.07	15
CDKN2A	4.44 × 10^−2^	−1.33	15
PDX1	1.81 × 10^−2^	−1.71	11
TP63	4.78 × 10^−3^	−2.01	25
GATA1	4.93 × 10^−5^	−2.03	21
Alpha catenin	1.36 × 10^−3^	−2.07	10
ROCK2	4.76 × 10^−3^	−2.24	6
PHB2	1.27 × 10^−4^	−2.24	4

# Genes Regulated: number of genes regulated.

**Table 6 ijms-22-07899-t006:** Top upstream regulators altered in female SPTB placentas.

Regulators	*p*-Value	Activation z-Score	# Genes Regulated
HNRNPK	2.94 × 10^−4^	——	3
INS	1.53 × 10^−3^	−1.09	4
AKT1	8.10 × 10^−3^	——	4
ZBTB16	1.61 × 10^−2^	——	3
PTEN	2.73 × 10^−2^	−1.00	5
IL15	3.11 × 10^−2^	——	4
ONECUT1	4.04 × 10^−2^	——	3
STAT5B	4.19 × 10^−2^	——	3
VEGF	4.75 × 10^−2^	−1.09	4

# Genes Regulated: number of genes regulated.

**Table 7 ijms-22-07899-t007:** Top master regulators altered in male SPTB placentas.

Master Regulators	*p*-Value	Activation z-Score	# Connected Regulators
MYB	8.33 × 10^−10^	3.76	6
TBK1	1.09 × 10^−10^	3.14	20
LGALS1	5.96 × 10^−10^	3.03	15
GAB2	7.01 × 10^−10^	2.71	23
NOX1	2.89 × 10^−11^	2.68	14
MAP3K8	5.54 × 10^−11^	2.32	34
PPP2CA	4.05 × 10^−9^	−1.51	34
BRCA1	8.78 × 10^−10^	−2.08	27
CCHCR1	1.75 × 10^−11^	−2.51	29
RBP1	4.24 × 10^−10^	−2.84	4
PPP2R2A	1.87 × 10^−8^	−3.03	12
DNAJA3	1.07 × 10^−9^	−3.16	11
MEN1	1.16 × 10^−8^	−3.25	8

# Connected Regulators: number of connected regulators.

**Table 8 ijms-22-07899-t008:** Top master regulators altered in female SPTB placentas.

Master Regulators	*p*-Value	Activation z-Score	# Connected Regulators
Fe2+	1.97 × 10^−2^	2.67	16
NFAT (family)	4.89 × 10^−3^	2.40	23
CAMKK2	1.60 × 10^−2^	2.12	5
BLVRA	7.49 × 10^−3^	2.07	23
MAPK13	5.11 × 10^−4^	2.04	26
TRERF1	3.82 × 10^−2^	−2.00	2
CERK	2.16 × 10^−2^	−2.50	13

# Connected Regulators: number of connected regulators.

**Table 9 ijms-22-07899-t009:** Metabolic pathways identified from the interactome network.

Metabolic Pathways Enriched within the Interactome Network	Number of Gene Changes (Inferred and Non-Inferred)	Gene Changes within Dataset	Metabolite Changes within Dataset	Number of Metabolites Changes (Inferred and Non-Inferred)
Androgen and estrogen biosynthesis and metabolism	98	CYP3A5, CYP2C18, MGST1, AKR1C2, SULT1E1	3beta-Hydroxyandrost-5-en-17-one 3-sulfate, Androst-5-ene-3beta,17beta-diol, Estrone	64
Arachidonic acid metabolism	114	HADHA, HADHB, CYP2C18, CYP3A5, PLA2G2A, PLA2G4A, ECHS1, MGST1, CYP4F3	15(S)-HETE, 5(S)-HETE	63
C21-steroid hormone bioshnthesis and metabolism	50	CYP2C18, CYP3A5, SULT1E1	Cholesterol, Cortisone	45
Fatty acid beta-oxidation and metabolism	57	CPT2, HADHA, HADHB, ECHS1	L-Palmitoylcarnitine	184
Fat-soluble vitamin metabolism	65	ALDH1A1, HADHA, HADHB, ECHS1, CYP4F3	No metabolites with significant difference	49
Fructose, galactose, and aminosugars metabolism	62	AKR1B1, NPL	N-Acetyl-D-glucosamine 1-phosphate, L-Glutamine, CMP-N-acetylneuraminate, N-Acetylneuraminate, N-Acetyl-D-glucosamine 6-phosphate	65
Glycerophospholipid metabolism	113	ADHFE1, AKR1B1, PLA2G2A, PLA2G4A, LIPA, ALDH7A1, CEL	L-Serine, sn-Glycerol 3-phosphate, Glycerone phosphate, Cholesterol, Ethanolamine phosphate, Choline phosphate	62
Glycine, serine, alanine, and threonine metabolism	61	DLD, BHMT2, TARS, GATM, CKMT1B	Glycine, L-Serine, L-Methionine, L-Threonine, Betaine aldehyde, Guanidinoacetate	59
Glycolysis and gluconeogenesis	80	ADHFE1, DLD, LDHB, ALDH7A1, PCK2	Phosphoenolpyruvate, Glycerone phosphate, Acetyl phosphate, 2-Phospho-D-glycerate	34
Glycosphingolipid metabolism	83	GALC, ASAH1	CMP, L-Serine, CMP-N-acetylneuraminate, Ethanolamine phosphate	116
Histidine and lysine metabolism	69	SUV39H2, ALDH7A1	L-2-Aminoadipate, L-Glutamate, Glycine, L-Lysine, beta-Alanine, Carnosine	50
Leukotriene metabolism	115	ADHFE1, HADHA, HADHB, CYP2C18, CYP3A5, ECHS1, MGST1, ALDH7A1, CYP4F3	Glycine	83
Linoleate metabolism	80	CYP2C18, CYP3A5, PLA2G2A, PLA2G4A	No metabolites with significant difference	16
Methionine and cysteine metabolism	51	MAT2A, LDHB, CBS	N-Acetylmethionine, L-Serine, L-Methionine, 3-Sulfino-L-alanine	41
Pentose phosphate pathway	30	SAT1, PGD	D-Sedoheptulose 7-phosphate, Glycerone phosphate	27
Phosphatidylinositol phosphate metabolism	98	MINPP1, PLCL2, OCRL, SYNJ2	myo-Inositol, Ethanolamine phosphate	45
Prostaglandin formation	50	MGST1	Prostaglandin E2	53
Purine metabolsim	274	ATP8A2, ATP5J, ATP5O, POLD2, HPRT1, ATP2B1, AMPD2, POLS	L-Aspartate, L-Glutamine, Urate	65
Pyrimidine metabolism	119	POLD2, UPB1, CAD, POLS	L-Aspartate, CMP, L-Glutamine, beta-Alanine	45
Squalene and cholesterol biosynthesis	26	SC4MOL, HMGCR, IDI1	Cholesterol	31
TCA cycle	26	DLD, IDH1, PCK2	Phosphoenolpyruvate	20
Tryptophan and tyrosine metabolism	169	FAH, ADHFE1, HADHA, HADHB, CYP2C18, CYP3A5, ECHS1, MGST1, CAT, ALDH7A1	Adenosine 3′,5′-bisphosphate, L-Phenylalanine, L-Tyrosine, Indole-3-acetate	157
Urea cycle and metabolism of arginine, proline, glutamate, aspartate, and asparagine	124	EPRS, GATM, CPS1, SAT1, ARG2, MGST1, ALDH7A1	L-Glutamate, Glycine, L-Lysine, L-Aspartate, L-Glutamine, beta-Alanine, L-Proline, L-Asparagine, 4-Aminobutanoate, trans-4-Hydroxy-L-proline, 5-Oxoproline, N1-Acetylspermine, O-Acetylcarnitine, N-Acetylputrescine, 4-Acetamidobutanoate, N1,N12-Diacetylspermine	108
Valine, leucine, and isoleucine degradation	54	HADHA, HADHB, ECHS1, ALDH7A1	L-Valine, L-Isoleucine, L-Leucine	43
Vitamin B metabolism	76	NMNAT1, PNPO, MTHFD1, NNMT, MTHFD1L, COASY	Pantothenate, FAD, L-Glutamate, Glycine, L-Lysine, L-Glutamine, L-Serine, Nicotinamide, 1-Methylnicotinamide, Pyridine-2,3-dicarboxylate	70
Xenobiotics metabolism	85	ADHFE1, CYP2C18, CYP3A5, AKR1C2, MGST1	No metabolites with significant difference	72

Genes or metabolites shown in red were upregulated in SPTB placentas, whereas those in green were downregulated.

## Data Availability

Sequence data have been deposited in NCBI’s Gene Expression Omnibus and are accessible through GEO Series accession number GSE174415.

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
