# Peer review of "Human Placental Transcriptome Reveals Critical Alterations in Inflammation and Energy Metabolism with Fetal Sex Differences in Spontaneous Preterm Birth"

_ijms, 2021, doi:10.3390/ijms22157899_

Round 1
Reviewer 1 Report
Lien et al manuscript is focusing on a highly relevant medical issue of preterm birth. The reviewer finds the study of high interest to the field. The strength of the study is the study design, careful selection of subjects, and the specific focus on minority population, highly affected population with high pre-term birth case numbers. The major clear difference found in gene expression between males and females is also of high importance and will be of interest to broader audience.
Minor issues: The manuscript result section itself could be slightly improved, if few titles (in line 206,318,329,347) were revised to be more clear, and results could be shortened and one summary sentence would be added to each subsection to summarize it and to help focusing attention on the major conclusions of each subsection.
Author Response
We thank the reviewer for the suggestions. The titles in the Result section have been modified and summary sentences have been added in the revised manuscript.
Reviewer 2 Report
The authors have presented a very interesting work, well written and with results that support the conclusions. Personally, as a physician who works in the field of obstetric and maternal-perinatal pathology, I have to congratulate the authors, I believe that the manuscript will be cited. Only small doubts arise, the sample size is limited. This point must be justified, I recommend that the authors make an adequate statistical justification. Again, I congratulate the authors.
Author Response
We thank the reviewer for the suggestion. We have included the power calculation that we performed when we designed this study. We used a false positive rate of 0.05 (two-sided) and power of 80% to target a 2-fold change, and the result indicates that 6 samples are sufficient to tell the differences. This information has been added to the Result section in the revised manuscript.